# Reversible Modulation of Motile Cilia by a Benzo[*e*][1,2,4]triazinone: A Potential Non-Hormonal Approach to Male Contraception

**DOI:** 10.3390/cells14100688

**Published:** 2025-05-09

**Authors:** Maria Chatzifrangkeskou, Alexandra Perdiou, Revekka Kreouzou, Georgia A. Zissimou, Dragos F. Flesariu, Panayiotis A. Koutentis, Paris A. Skourides

**Affiliations:** 1Department of Biological Sciences, University of Cyprus, P.O. Box 20537, 1678 Nicosia, Cyprus; chatzifrangkeskou.maria@ucy.ac.cy (M.C.); al.perdiou@gmail.com (A.P.); kreouzou.revekka@ucy.ac.cy (R.K.); 2Department of Chemistry, University of Cyprus, P.O. Box 20537, 1678 Nicosia, Cyprus; zissimou.georgia@ucy.ac.cy (G.A.Z.); dflesa01@ucy.ac.cy (D.F.F.)

**Keywords:** motile cilia, deciliation, *Xenopus laevis*, transition zone

## Abstract

Motile cilia play essential roles in various physiological processes including fluid flow generation and sperm motility. In this study, we identified 1,3-diphenyl-6-(4-phenylpiperazin-1-yl)benzo[*e*][1,2,4]triazin-7(1*H*)-one as a potent and reversible modulator of ciliary function using the *Xenopus laevis* model. This benzotriazinone derivative inhibits ciliary-driven fluid flow by inducing cilia detachment without causing toxicity in developing embryos. Unlike traditional deciliation agents that rely on calcium signaling, this compound induces cilia loss through a shear stress-driven mechanism at the transition zone, without disrupting tissue morphology or the apical actin network. Importantly, it also induces flagellar loss and impairs sperm motility at picomolar concentrations. Our findings highlight the potential of this 6-(4-phenylpiperazin-1-yl)-substituted benzotriazinone as a non-hormonal male contraceptive and underscore a novel mechanism of cilia modulation that may have broader implications for the treatment of cilia-related disorders.

## 1. Introduction

Cilia are essential, hair-like structures on eukaryotic cells that regulate fluid movement, sensory perception, and cell signaling. Multiciliated cells (MCCs) found in tissues, such as the respiratory tract, oviducts, and embryonic epidermis, facilitate coordinated fluid transport. In reproduction, motile cilia drive flagellar beating, enabling sperm motility and fertilization. Given their critical roles, disruptions in cilia function contribute to diseases such as primary ciliary dyskinesia, infertility, and respiratory disorders. Understanding the regulation of ciliogenesis is crucial for addressing these conditions.

Cilia can be lost through two mechanisms: resorption and deciliation [1]. Resorption is a slow, controlled process involving changes in intraflagellar transport and microtubule modifications [2,3]. In contrast, deciliation is a rapid, stress-induced process triggered by environmental factors such as toxins or hormonal signals, resulting in microtubule severing at the transition zone and membrane sealing [1,4,5,6]. This process is known as autotomy and is conserved across various ciliated tissues [1]. While calcium signaling is known to trigger deciliation, the exact proteins involved remain unclear.

Deflagellation serves as a critical tool for studying cilia regeneration and microtubule severing, offering insights into diseases such as polycystic kidney disease, retinal degeneration, and situs inversus.

*Xenopus laevis* is an ideal model for studying cilia biology and assessing the effects of pharmacological agents on multiciliated cells. Its well-characterized mucociliary epithelium provides a powerful platform for visualizing the motile cilia structure and function in response to drug treatments under controlled conditions.

1,3-Diphenylbenzo[*e*][1,2,4]triazin-7-one (**1**) (Appendix A) [7] is a versatile heterocyclic scaffold [8,9,10] with interesting biological activities including antiproliferative effects and thioredoxin reductase inhibition [11]. Recent studies have also highlighted its potential therapeutic applications in Alzheimer’s disease [12]. However, the effects of benzotriazinones on ciliary function remain unexplored.

In this study, we screened a small library of benzotriazinones and identified 1,3-diphenyl-6-(4-phenylpiperazin-1-yl)benzo[*e*][1,2,4]triazin-7(1*H*)-one (10) (Table 1) as a specific derivative with pronounced effects on ciliary dynamics. Using the *Xenopus laevis* model, we investigated its impact on ciliary function, aiming to elucidate its mechanism of action, particularly its potential interactions with ciliary microtubules or motor proteins, and evaluated its promise as both a research tool for studying cilia-related disorders and a non-hormonal contraceptive.

## 2. Materials and Methods

### 2.1. General Methods and Materials

All chemicals were commercially available except those whose synthesis is described. Volatiles were removed under reduced pressure. Solvents were distilled over CaH_2_ before use. Reactions were protected from moisture with CaCl_2_ drying tubes. Reaction mixtures and column eluents were monitored by TLC using commercial glass-backed thin layer chromatography (TLC) plates (Merck Kieselgel 60 F_254_, Darmstadt, Germany). The plates were observed under UV light at 254 and 365 nm. The technique of dry flash chromatography was used throughout for all non-TLC scale chromatographic separations using Merck Silica Gel 60 (<0.063 mm). Melting points were determined using a PolyTherm-A, Wagner & Munz, Koefler—Hotstage Microscope apparatus (Wagner & Munz, Munich, Germany). Solvents used for recrystallization are indicated after the melting point. UV–Vis spectra were obtained using a Shimadzu UV-1900 (Shimadzu, Kyoto, Japan), and inflections are identified by the abbreviation “inf”. IR spectra were recorded on a Shimadzu FTIR-NIR Prestige-21 spectrometer (Shimadzu, Kyoto, Japan) with a Pike *Miracle* Ge ATR accessory (Pike Miracle, Madison, WI, USA), and strong, medium, and weak peaks are represented by s, m and w, respectively. ^1^H and ^13^C NMR spectra were recorded on a Bruker Avance 500 (at 500 and 125 MHz, respectively) (Bruker, Billerica, MA, USA). Deuterated solvents were used for homonuclear lock, and the signals were referenced to the deuterated solvent peaks. MALDI-TOF mass spectra were recorded on a Bruker Autoflex III Smartbeam instrument (Bruker, Billerica, MA, USA). 1,3-Diphenylbenzo[*e*][1,2,4]triazin-7(1*H*)-one (**1**) [1], 1-phenyl-3-(trifluoromethyl)benzo[*e*][1,2,4]triazin-7(1*H*)-one (**2**) [13], 6-amino-1,3-diphenylbenzo[*e*][1,2,4]triazin-7(1*H*)-one (**3**) [9], 6-(ethylamino)-1,3-diphenylbenzo[*e*][1,2,4]triazin-7(1*H*)-one (**4**) [9], 6-(2-methylpiperidin-1-yl)-1,3-diphenylbenzo[*e*][1,2,4]triazin-7(1*H*)-one (**5**) [12], 6-(3-methylpiperidin-1-yl)-1,3-diphenylbenzo[*e*][1,2,4]triazin-7(1*H*)-one (**6**) [12], 6-(4-methylpiperidin-1-yl)-1,3-diphenylbenzo[*e*][1,2,4]triazin-7(1*H*)-one (**7**) [12], 6-morpholino-1,3-diphenylbenzo[*e*][1,2,4]triazin-7(1*H*)-one (**8**) [9], 6-(4-methylpiperazin-1-yl)-1,3-diphenylbenzo[*e*][1,2,4]triazin-7(1*H*)-one (**9**) [9], 1,3-diphenyl-6-(4-phenylpiperazin-1-yl)benzo[*e*][1,2,4]triazin-7(1*H*)-one (**10**) [9], 1,3-diphenyl-6-(4-phenylpiperidin-1-yl)benzo[*e*][1,2,4]triazin-7(1*H*)-one (**11**) [9], 6-[(2-aminophenyl)amino]-1,3-diphenylbenzo[*e*][1,2,4]triazin-7(1*H*)-one (**12**) [9], 6-{[3-(dimethylamino)propyl](methyl)amino}-1,3-diphenylbenzo[*e*][1,2,4]triazin-7(1*H*)-one (**13**) [12], 6-methoxy-1,3-diphenylbenzo[*e*][1,2,4]triazin-7(1*H*)-one (**14**) [14], 1,3-diphenyl-6-(phenylthio)benzo[*e*][1,2,4]triazin-7(1*H*)-one (**15**) [14], 1,3-diphenyl-6-(thiophen-2-yl)benzo[*e*][1,2,4]triazin-7(1*H*)-one (**16**) [9], 2-phenyl-5-(pyrrolidin-1-yl)-6*H*-[1,2,4]triazino[5,6,1-*jk*]carbazol-6-one (**17**) [12], 2-phenyl-5-(phenylthio)-6*H*-[1,2,4]triazino[5,6,1-*jk*]carbazol-6-one (**18**) [14], 1,3,8-triphenylbenzo[*e*][1,2,4]triazin-7(1*H*)-one (**19**) [9], 8-(3-methoxyphenyl)-1,3-diphenylbenzo[*e*][1,2,4]triazin-7(1*H*)-one (**20**) [9], 8-(4-chlorophenyl)-1,3-diphenylbenzo[*e*][1,2,4]triazin-7(1*H*)-one (**21**) [9], 8-(fur-2-yl)-1,3-diphenylbenzo[*e*][1,2,4]triazin-7(1*H*)-one (**22**) and 2-(4-phenylpiperazin-1-yl)cyclohexa-2,5-diene-1,4-dione (**25**) [15] were prepared according to the literature.

2,5-Bis(4-phenylpiperazin-1-yl)cyclohexa-2,5-diene-1,4-dione (**26**). To a stirred solution of 1,4-benzoquinone (**23**) (108 mg, 1.0 mmol) in CH_2_Cl_2_ (5 mL) at *ca*. 20 °C was added dropwise 1-phenylpiperazine (**24**) (325 mg, 2.0 mmol). The reaction, which turned red immediately, was stirred for 10 min at *ca*. 20 °C. Upon completion, volatiles were removed *in vacuo* and the solid residue was recrystallized to give the 2,5-Bis(4-phenylpiperazin-1-yl)cyclohexa-2,5-diene-1,4-dione (**26)** as red plates (230 mg, 54%): mp (hotstage) 220–222 °C (EtOH), decomposes immediately; Anal. Calcd for C_26_H_28_N_4_O_2_: C, 72.87; H, 6.59; N, 13.07. Found: C, 72.75; H, 6.72; N, 12.98%; *λ*_max_(DCM)/nm 252 (log *ε* 4.77), 368 (4.50), 534 (3.02); *ν*_max_/cm^−1^ 3039w (Ar CH), 2947w (CH), 2831 (CH), 1636s (C=O), 1597m, 1566s, 1497m, 1443m, 1412w, 1389w, 1373m, 1319w, 1304m, 1219s, 1173w, 1134m, 1034m, 941s, 826s, 756s; ^1^H NMR (500 MHz, CDCl_3_) *δ*_H_ 7.29 (dd, *J* 8.8, 7.2, 2H), 6.90 (dd, *J* 14.5, 7.4, 3H), 5.61 (s, 1H), 3.76 (t, *J* 5.3, 4H), 3.33 (t, *J* 5.3, 4H); ^13^C NMR (126 MHz, CDCl_3_) *δ*_C_ 182.7, 152.4, 150.7, 129.4, 120.3, 116.1, 106.6, 48.8, 48.7; *m*/*z* (MALDI-TOF) 429 (MH^+^, 26%), 428 (M^+^, 59), 427 (40), 426 (100). The ^1^H/^13^C NMRs of compound **26** are provided in Appendix A.

### 2.2. Stock Solutions

Compounds were dissolved in DMSO to make a stock solution with a 1 mM concentration and then diluted in 0.1× Marc’s Modified Ringer’s (MMR) for the indicated working concentrations.

### 2.3. Xenopus laevis Embryo Manipulation and Microinjections

Male frogs were sacrificed in 0.06% benzocaine for 20–30 min and then dissected. Testes were preserved at 4 °C in 10% fetal calf serum (FCS), L-glutamine, and 50 mg/mL gentamicin, diluted in L-15 medium. Female adult *Xenopus laevis* were induced to ovulation through the injection of glycoprotein hormone human chorionic gonadotropin (hCG). The administration was performed 15 h before the collection of eggs, with 600–700 IU of the protein. The collected eggs were put in a Petri dish containing 0.33× Marc’s Modified Ringer’s (MMR) solution. Fertilization of the eggs was performed in vitro using sperm cells from the dissected testes. Then, the embryos were de-jellied in 1.8% L-cysteine (pH 7.8) in 0.33× MMR, washed in 0.33× MMR, and raised in 0.1× MMR. Microinjections were performed in 4% Ficoll in 0.33× MMR. The injections were performed using a glass capillary pulled needle, forceps, a Singer Instruments MK1 micromanipulator, and a Harvard Apparatus pressure injector. For all experiments, injections were made into the ventral blastomeres of 4-cell or 8-cell stage embryos to target the epidermis. A total of 150 pg of Geco-red mRNA, 100 pg memCherry mRNA, 50 pg of B9D1 DNA, and 60 pg centrin were injected into each blastomere. mEmerald B9D1, RFP-Centrin, and mEmerald MKS1 were previously used [16]. The Geco-red construct was purchased from Addgene and subcloned into the CS108 plasmid for in vitro transcription.

### 2.4. Chemical Deciliation of Embryos

Deciliation was performed as previously described [17]. Briefly, the embryos were incubated in deciliation buffer (75 mM CaCl_2_, 0.02% NP40 in 0.1 MMR) at room temperature (RT) for 1 min. To remove the cellular debris, samples were centrifuged at 1500× *g* for 2 min, and the supernatants were transferred to 0.1% poly-L-lysine coated coverslips and immediately fixed.

### 2.5. Immunofluorescence

Embryos were fixed in 1× MEMFA (10×: 1 M MOPS, 20 mM EGTA, 10 mM MgSO_4_, 38% formaldehyde) for 2 h at RT. Then, the embryos were permeabilized in 1 × PBS (phosphate buffered saline) + 0.5% Triton X-100 + 1% DMSO) at RT and blocked in PBDT + 1% donkey serum for 1 h at RT. For the centrin antibody (Table 2), methanol fixation was performed overnight at −20 °C. Primary antibodies were added in blocking solution, and the embryos were incubated overnight at 4 °C. The next day, the embryos were washed in PBDT and incubated with secondary antibodies for 1 h at RT (Table 2). Starved NIH3T3 cells were fixed with 4% paraformaldehyde for 10 min, permeabilized with 0.3% Triton-X 100 in 1× PBS for 10 min and blocked in 10% donkey serum for 30 min at RT. Cells were then incubated with the primary antibodies at RT for 1 h, followed by three washes with 1× PBS. Subsequently, cells were incubated with secondary antibodies for 1 h at RT. Samples were mounted on coverslips with Prolong diamond (Invitrogen, Thermo Fisher Scientific, Waltham, MA, USA). Samples were visualized using a Zeiss LSM 900 laser scanning confocal microscope with an Airyscan imaging system (Carl Zeiss Microscopy GmbH, Jena, Germany)

### 2.6. Assessment of Cilia Generated Fluid Flow

The fluid flow generated through the *Xenopus* mucociliary epithelium was assessed by tracking the velocity of the fluorescent microspheres. The embryos were anesthetized using 0.01% benzocaine diluted in 0.1 MMR. Fluorescent microspheres (Life Technologies, Carlsbad, CA, USA, #F8816) were added to the media. Flow video recordings were performed on a Zeiss Axio Imager Z1 microscope equipped with Zeiss Axiocam MR3, using Axiovision software 4.8.2. Time-lapsed movies were recorded for 10 s (camera binned 3 × 3 monochrome and recording at maximum speed). Tracking and quantification of flow velocities was performed using the IMARIS software v.9.0.1 64x.

### 2.7. Quantifications and Statistics

Statistical analysis was performed using the two-tailed unpaired *t*-test with the 95% confidence interval. Data are represented the as mean ± SEM. All experiments were independently repeated at least three times.

Hoechst 33342 nuclear stain was used to count the percentage of ciliated cells. The cilium length was quantified using Zeiss ZEN 3.7 software.

## 3. Results

### 3.1. 1,3-Diphenyl-6-(4-phenylpiperazin-1-yl)benzo[e][1,2,4]triazin-7(1H)-one *(**10**)* Robustly and Reversibly Inhibits Fluid Flow Generation

Extensive screening using our *Xenopus*-based assay [18] revealed that selected benzotriazinyl derivatives can suppress ciliary flow. Based on these findings, a targeted screen was performed on a small in-house library of benzotriazinyl derivatives (Table 1). Stage 30 *Xenopus laevis* embryos were exposed overnight to increasing concentrations of the tested compounds and assessed for survival (Figure 1A). The highest concentration that allowed for embryo survival was selected for further testing. The assessment of possible fluid flow alterations in the *Xenopus* epidermis resulting from compound incubation was carried out through fluorescent bead visualization of the fluid flow. Specifically, the fluid flow velocity was calculated by tracking fluorescent microspheres in the media surrounding the embryos [17]. To assess changes in mucociliary clearance, flow assays were performed before and 1 h after the introduction of the selected compound in the media.

Among the compounds tested, 1,3-diphenyl-6-(4-phenylpiperazin-1-yl)benzo[*e*][1,2,4]triazin-7(1*H*)-one (**10**) demonstrated significant potency in inhibiting the fluid flow in *Xenopus laevis* embryos treated with a concentration of 50 μM (Figure 1B, Appendix A). Based on this effect, we then performed a dose–response analysis using lower concentrations of benzotriazinone **10** (0.1–5 nM) for 1 h, which showed a dose-dependent reduction in fluid flow (Figure 1C). The data revealed an IC_50_ as low as 0.1 nM, highlighting the compound’s high potency. These results confirm a strong dose-dependent inhibition of fluid flow generation.

To evaluate the potential toxicity of benzotriazinone **10**, embryos were treated at an early developmental stage (Stage 11). Notably, no lethality or overt toxic effects were observed, even at concentrations that strongly inhibited flow generation, suggesting that compound **10** did not adversely affect embryonic development (Appendix A).

### 3.2. 1,3-Diphenyl-6-(4-phenylpiperazin-1-yl)benzo[e][1,2,4]triazin-7(1H)-one *(**10**)* Promotes Shear Stress-Driven Deciliation

We investigated the mechanism by which benzotriazinone **10** modulates fluid flow and affects ciliary function. Tadpoles treated with benzotriazinone **10** for 1 h exhibited no overall changes in mucociliary epithelium integrity or cytoskeletal elements, as indicated by intact β-catenin (S3a), a key cell–cell adhesion protein essential for epithelial integrity [19,20] and stable actin networks (S3b). However, immunostaining with an acetylated tubulin antibody revealed a marked reduction in the cilia density of epidermal MCCs (Figure 2A), suggesting that benzotriazinone **10** specifically targets ciliary function without compromising tissue morphology.

To further explore its temporal effects, we performed time-course immunostaining (Figure 2B). Ciliary disruption was evident within 30 s of treatment, with a significant reduction in cilia density observed at 10 min. These findings aligned with our fluid flow assays, where a sharp decline in velocity was detected within 30 s. Notably, these effects were reversible; cilia regeneration began within 1 h of drug removal, with complete recovery by 3 h, indicating that benzotriazinone **10** did not cause permanent ciliary damage (Figure 2C).

To assess specificity, we examined its effects on primary cilia in the NIH3T3 cells. After serum starvation-induced ciliogenesis, benzotriazinone **10** treatment did not alter the number or length of the primary cilia compared with the controls (Appendix A), confirming its selective impact on motile cilia.

To better understand the mechanism by which benzotriazinone **10** induces ciliary loss, we examined whether the drug caused cilia retraction or shedding. We thus isolated media from treated tadpoles and examined it for the presence of cilia. As shown, the media after treatment contained numerous cilia, suggesting that benzotriazinone **10** induces cilia shedding rather than retraction (Figure 3A). To confirm cilia shedding, we carried out time-lapse imaging of benzotriazinone **10**-treated embryos stained with a plasma membrane dye and confirmed rapid cilia detachment (Figure 3B, Appendix A). As a positive control, we used calcium shock (deciliation buffer), which also resulted in cilia detachment. However, in embryos treated with benzotriazinone **10**, the detached cilia exhibited a distinct curled morphology, whereas those shed by calcium shock maintained a more linear form. This suggests that benzotriazinone **10** induces cilia autotomy via a mechanism that likely affects the axonemal structure or the transition zone differently from calcium shock. The curling might reflect differential impacts on microtubule stability or structural reorganization, hinting at a unique underlying molecular mechanism. To confirm that benzotriazinone **10** did not induce cilia loss through calcium signaling, we monitored intracellular Ca^2+^ levels using Geco-red, a calcium ion indicator, in combination with membrane GFP (memGFP) to visualize the cilia. Unlike the deciliation buffer, which increased the Ca^2+^ levels, benzotriazinone **10** did not alter the intracellular Ca^2+^ concentrations (Appendix A). This confirms that benzotriazinone **10**’s mechanism of action is distinct from the calcium-mediated pathways. Interestingly, blocking ciliary motility with NiCl_2_ prior to benzotriazinone **10** treatment prevented cilia loss (Figure 3C), suggesting that the shear stress generated by ciliary beating plays a role in the deciliating effect of benzotriazinone **10**.

### 3.3. 1,3-Diphenyl-6-(4-phenylpiperazin-1-yl)benzo[e][1,2,4]triazin-7(1H)-one *(**10**)* Induces Cilia Shedding at the Distal Region of the Transition Zone

To determine the precise site of cilium severing upon benzotriazinone **10** treatment, we conducted post-treatment immunofluorescence against centrin 1 to label the basal bodies, the nucleation site of the cilium. As shown, the basal bodies remained intact in the *Xenopus* epidermis after benzotriazinone **10** treatment (Appendix A). We then investigated whether benzotriazinone **10**-mediated excision occurred at the same site as the commonly used deciliation buffer using the proximal transition zone protein B9D1 (Figure 4A). While B9D1 was absent from MCCs treated with the deciliation buffer, it was present in the benzotriazinone **10**-treated embryos, suggesting that cilium excision occurs at a distal location compared with calcium-mediated deciliation (Figure 4A and Appendix A). We went on to microinject embryos with fluorescently labeled MKS1, a distal marker of the transition zone [16]. MKS1 was not detected in the benzotriazinone **10**-treated embryos, indicating that benzotriazinone **10** induces cilia autotomy at the transition zone, retaining the proximal region while the distal region is removed along with the ciliary axonemes (Figure 4B). This differential shedding site underscores a unique mechanism of cilia loss promoted by benzotriazinone **10**.

### 3.4. Structural Requirements for 1,3-Diphenyl-6-(4-phenylpiperazin-1-yl)benzo[e][1,2,4]triazin-7(1H)-one *(**10**)* Activity

To identify the essential structural features of the 6-(4-phenylpiperazin-1-yl)-substituted benzotriazinone **10**, we conducted a structure–activity relationship (SAR) study. This investigation focused on evaluating the contributions of individual molecular fragments to the compound’s biological activity.

Neither 1,4-benzoquinone (**23**) nor 1-phenylpiperazine (**24**) alone inhibited ciliary flow (Figure 5A), indicating that a quinone core or a piperazine group alone is insufficient for activity. Similarly, larger fragments that combined the piperazinyl and quinone groups but lacked the triazine moiety, such as 2-(4-phenylpiperazin-1-yl)cyclohexa-2,5-diene-1,4-dione (**25**) and 2,3-bis(4-phenylpiperazin-1-yl)cyclo-hexa-2,5-diene-1,4-dione (**26**), also failed to inhibit ciliary flow. In contrast, 1,3-diphenyl-benzo[*e*][1,2,4]triazine-7(1*H*)-one (**1**), which represents the core heterocyclic scaffold of benzotriazinone **10**, displayed partial inhibitory effects (Table 1, entry 1). This finding underscores the importance of the benzotriazinone core for activity.

To determine the effect of the C3 substituent on the benzotriazinone core, two benzotriazinones were compared: the 3-phenyl-substituted benzotriazinone **1** (Table 1, entry 1) and the 3-CF_3_–substituted benzotriazinone **2** (Table 1, entry 2). The former exhibited greater activity in terms of the track speed mean, suggesting that the C3-phenyl plays a significant role, likely due to additional π–π stacking interactions or improved electronic properties that enhance binding to the biological target.

The addition of 1-phenylpiperazine onto the benzotriazinone scaffold **1**, yielding the 6-(4-phenylpiperazin-1-yl)-substituted derivative **10** (Table 1, entry 11), significantly enhanced potency. This increase in activity can be attributed to optimized interactions with the biological target, emphasizing the importance of both the triazine core and the precise placement of substituents.

Interestingly, the structurally similar 6-(4-phenylpiperidin-1-yl)-substituted benzotriazinone **11** (Table 1, entry 12) elicited similar ciliary phenotypes but with reduced efficacy in fluid flow inhibition compared with the phenylpiperazine analog (Figure 5B,C). The reduced activity of the piperidinyl derivative may stem from its single nitrogen atom, which limits its ability to participate in hydrogen bonding, protonation, or other electronic interactions with the target. In contrast, the piperazine ring, with its two nitrogen atoms, is a versatile “privileged structure” in medicinal chemistry, known for its ability to form multiple interactions with biological targets [21].

The 4-methylpiperazin-1-yl derivative **9** (Table 1, entry 10) exhibited significantly reduced activity compared with the 4-phenylpiperazin-1-yl analog **10**, further underscoring the critical role of the phenyl group at the 4-position of the piperazine ring. The phenyl group likely enhances the binding affinity or efficacy through π–π stacking, hydrophobic interactions, or other aromatic interactions with the target.

These findings highlight the essential structural features of benzotriazinone **10** for biological activity: the benzotriazinone core, the piperazine moiety, and the phenyl groups at the 3-position of the benzotriazinone and the 4-position of the piperazine. Together, these moieties optimize target interactions, making them key pharmacophores for further optimization.

### 3.5. 6-(4-Phenylpiperazin-1-yl)-Substituted Benzotriazinone ***10*** Inhibits Sperm Motility by Inducing Flagellar Loss

Given that sperm flagella and motile cilia share components and motility machinery, we investigated whether benzotriazinone **10** affected sperm function. Using sperm from mature *Xenopus laevis* males, we found that incubation with 0.1 nM benzotriazinone **10** induced immediate paralysis, severely impairing motility (Figure 6A). Further analysis via immunofluorescence, using acetylated tubulin to label the sperm flagellum, Hoechst 33342 dye to stain the nucleus, and a dye to mark the plasma membrane, revealed that the benzotriazinone **10**-treated sperm cells rapidly lost their flagellum (Figure 6B). This striking flagellar detachment suggests that benzotriazinone **10** disrupts flagellar integrity through a mechanism dependent on active motility, as further explored in the Discussion. These findings highlight the potential of benzotriazinone **10** as a non-hormonal contraceptive agent.

## 4. Discussion

Our study established 6-(4-phenylpiperazin-1-yl)-substituted benzotriazinone **10** as a reversible inhibitor of motile cilia, acting via a unique shear stress-driven deciliation mechanism that spares intracellular Ca^2+^ signaling and tissue architecture. By inducing rapid and reversible cilia loss in *Xenopus* embryos and triggering flagellar detachment in sperm, benzotriazinone **10** represents a promising non-hormonal strategy for male contraception.

Deciliation has been widely studied in both physiological processes and disease contexts. Traditional deciliation agents, such as Ca^2+^ shock, induce cilia loss through calcium-mediated pathways, leading to the rapid shedding of cilia from multiciliated cells. Our study demonstrated that benzotriazinone **10** promotes cilia shedding through a mechanism that does not involve a rise in the intracellular calcium levels, as evidenced by the absence of Ca^2+^ signaling during treatment. Instead, benzotriazinone **10** induces cilia loss at the distal region of the transition zone, a site distinct from the excision site seen with Ca^2+^ shock treatment [22]. This suggests that benzotriazinone **10** operates via a unique, shear stress-driven mechanism that requires active ciliary motility, such as NiCl_2_ pre-treatment, which inhibits cilia beating, prevented cilia shedding (Figure 3C). A major open and interesting question remains the precise molecular mechanism underlying benzotriazinone **10**–mediated cilia shedding. Identifying the direct molecular targets of the compound and dissecting the downstream pathways involved in this shear stress-dependent process will be key areas for future investigation.

This reversibility of the effects of benzotriazinone **10** indicates that the drug transiently disrupts the ciliary dynamics without causing irreversible damage to the ciliary machinery. This characteristic is advantageous for applications requiring temporary cilia inhibition, such as in male contraception, where reversible effects on sperm motility are desirable. The inhibition of sperm motility by benzotriazinone **10** is particularly significant, given the structural and functional similarities between the flagella and motile cilia. Traditional contraceptives include hormonal methods, such as levonorgestrel and ethinylestradiol, which prevent ovulation, fertilization, or implantation or non-hormonal methods, such as spermicides like nonoxynol-9, which disrupt sperm membranes or immobilize sperm cells [23]. In contrast, benzotriazinone **10** acts at picomolar concentrations, inducing rapid and irreversible flagellar detachment.

While the rapid action and low effective concentration of benzotriazinone **10** are promising, the potential for off-target effects on other ciliated tissues, such as those in the respiratory tract, must be carefully evaluated. Further research in mammalian models will be essential to assess the safety and efficacy of the drug in more complex biological systems, including uterine ciliary cells, where motility plays a key role in fertility. Additionally, developing targeted delivery systems, such as nanoparticles or tissue-specific ligands, could help confine the drug’s action to the reproductive system, minimizing off-target effects.

In conclusion, this study provides strong evidence for the efficacy of benzotriazinone **10** in rapidly inducing deciliation and inhibiting ciliary function at picomolar concentrations. While these findings are promising for the development of a novel contraceptive, careful consideration of the drug’s specificity and safety profile is essential for its future application.

## Figures and Tables

**Figure 1 cells-14-00688-f001:**
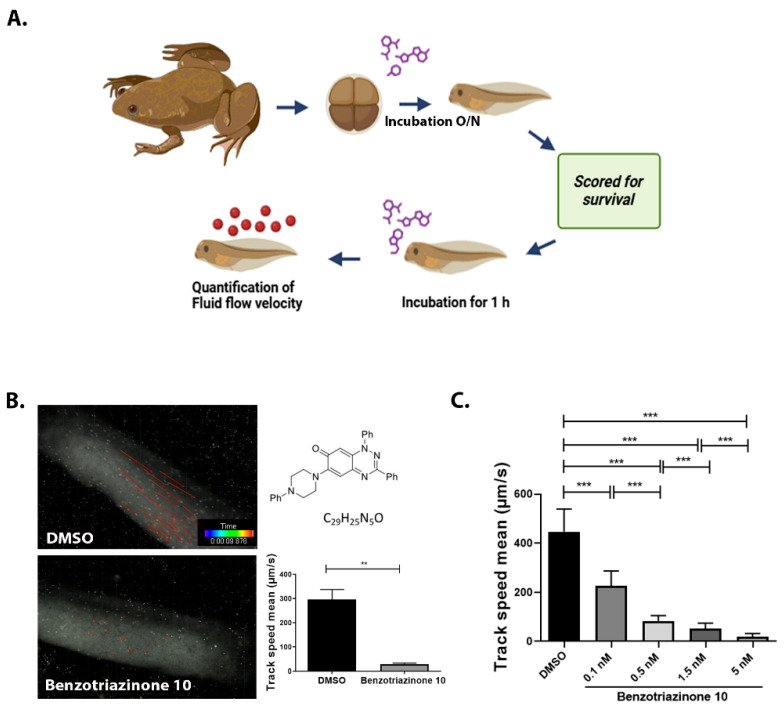
Benzotriazinone **10** robustly and reversibly inhibits fluid flow generation. (**A**) Schematic diagram illustrating the experimental workflow. *Xenopus* embryos were treated with chemical compounds overnight, followed by the assessment of survival scoring. Ciliary-driven fluid flow was assessed using fluorescent beads after 1 h of compound incubation. (**B**) Representative images of fluorescent bead tracking in embryos treated with DMSO (control) or benzotriazinone **10** (50 μΜ). Tracks indicate bead movement driven by ciliary motion (*n* = 4 embryos, ** *p* < 0.01). The chemical structure of benzotriazinone **10** is shown. Quantification of bead track speed (µm/s) demonstrated a significant reduction in fluid flow velocity in the benzotriazinone **10**-treated embryos compared with the controls. (**C**) Dose-dependent inhibition of fluid flow by benzotriazinone **10** (*n* = 4 embryos, *** *p* < 0.001). Fluid flow decreased significantly with increasing benzotriazinone **10** concentrations.

**Figure 2 cells-14-00688-f002:**
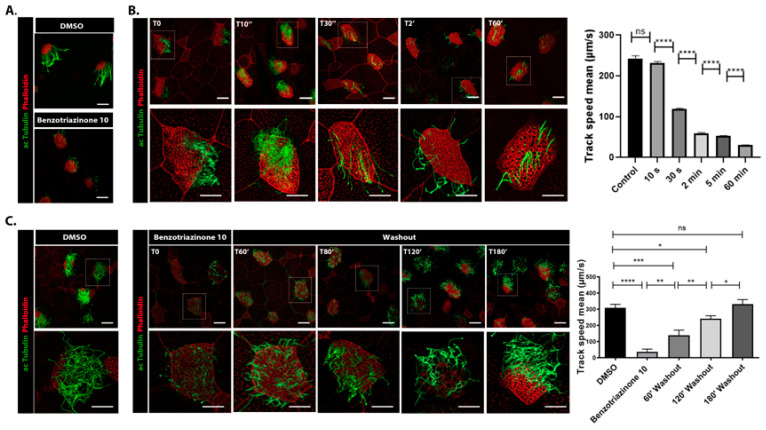
Benzotriazinone **10** induces cilia shedding in *Xenopus* embryos. (**A**) Representative confocal images showing acetylated α-tubulin and phalloidin in epidermal MCCs treated with DMSO (control) or 0.5 nM benzotriazinone **10**. Benzotriazinone **10** treatment led to complete cilia detachment within minutes, without disrupting the apical actin network. Scale bars: 10 µm. (**B**) Time-course analysis of cilia shedding in response to 0.5 nM benzotriazinone **10**. Confocal images showed acetylated α-tubulin and phalloidin at various time points following benzotriazinone **10** treatment. The bottom panels represent magnified insets marked by white squares. Quantification of the bead track speed showed a significant reduction in the fluid flow velocity, correlating with cilia loss (*n* = 3 embryos, **** *p* < 0.0001, ns not significant). Scale bars: 10 µm. (**C**) Reversibility of benzotriazinone **10**-induced cilia detachment. MCCs were treated with benzotriazinone **10** for 60 min, followed by a drug washout and recovery at 60, 80, 120, and 180 min (*n* = 4 embryos, * *p* < 0.05, ** *p* < 0.01, *** *p* < 0.001, **** *p* < 0.0001). The bottom panels represent magnified insets marked by white squares. Quantification of the bead speed demonstrated a statistically significant recovery over time. Scale bars: 10 µm.

**Figure 3 cells-14-00688-f003:**
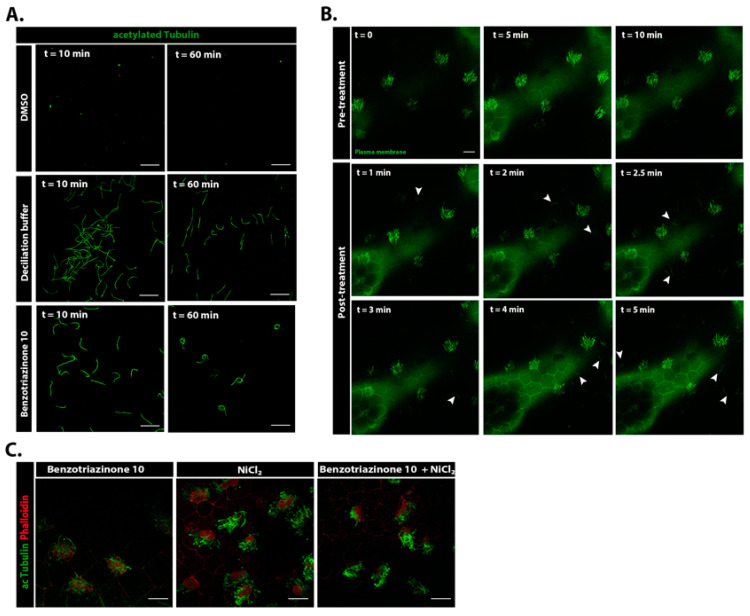
Benzotriazinone **10** promotes shear stress-driven deciliation. (**A**) Isolated cilia on poly-L-lysine-coated cover glass following deciliation buffer or 0.5 nM benzotriazinone **10**-treatment of the embryos. Acetylated α-tubulin was used to label the ciliary axonemes. Scale bar: 10 µm. (**B**) Stills from a time-lapse movie showing MCCs stained with a plasma membrane dye. Arrowheads show the severed cilia. Scale bar: 20 µm. (**C**) Immunostaining of embryos treated with either benzotriazinone **10** or NiCl_2_ or both. Inhibition of ciliary motility abrogated the effect of benzotriazinone **10**. Scale bar: 20 µm.

**Figure 4 cells-14-00688-f004:**
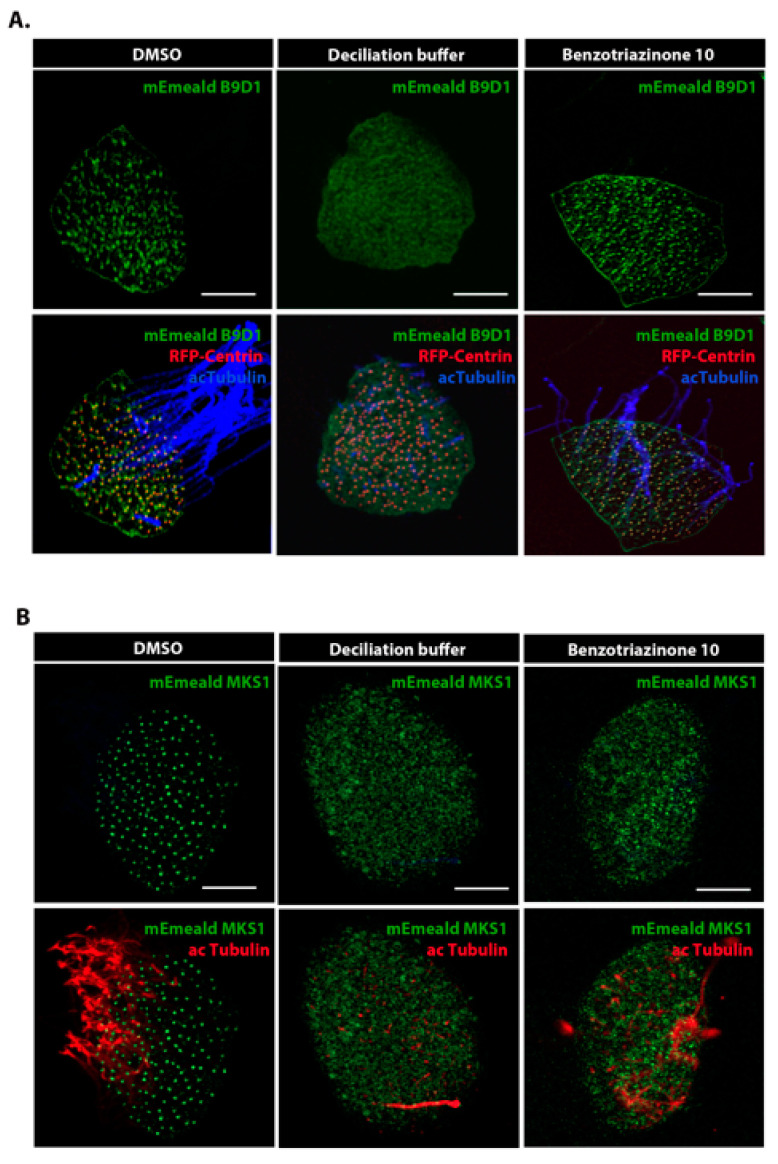
Benzotriazinone **10** induces cilia shedding at the distal region of the transition zone. (**A**) Representative Z-stack images of an epidermal MCC expressing mEmerald-tagged B9D1 and RFP-Centrin to mark the basal bodies. Acetylated α-tubulin was used to label the cilia. Treatment with 0.5 nM benzotriazinone **10** showed that the transition zone marker B9D1 was preserved. Scale bar: 10 µm. (**B**) Representative Z-stack images of a MCC expressing mEmerald-tagged MKS1 and RFP-Centrin. Benzotriazinone **10** loss of the transition zone marker MKS1. Scale bar: 10 µm.

**Figure 5 cells-14-00688-f005:**
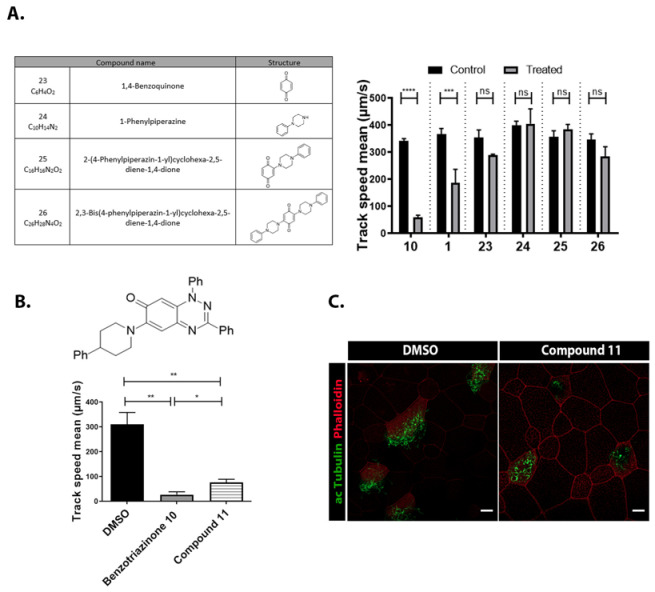
Structure–activity relationship analysis of benzotriazinone **10**. (**A**) Summary of the chemical structures of the benzotriazinone **10** analogs and their effects on ciliary flow. The graph shows the speed of fluorescent beads in *Xenopus* embryos treated with the indicated compounds or DMSO (*n* = 3 embryos, *** *p* < 0.001, **** *p* < 0.0001, ns, not significant). (**B**) Quantification of fluid flow in embryos treated with benzotriazinone **10** or its analog 11 compared with the DMSO-treated controls (*n* = 4 embryos, * *p* < 0.05, ** *p* < 0.01). (**C**) Representative confocal images of *Xenopus* epidermal MCCs treated with DMSO or analog 11 stained for acetylated tubulin and phalloidin. Scale bar: 10 µm.

**Figure 6 cells-14-00688-f006:**
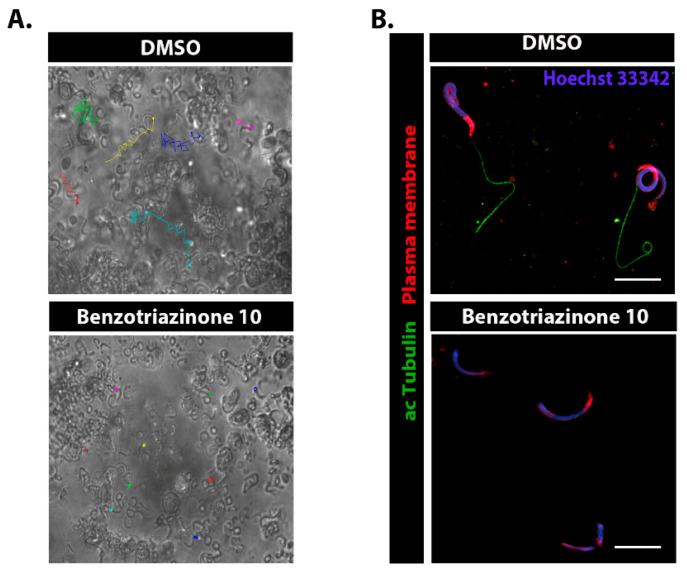
Effects of benzotriazinone **10** on sperm motility and morphology. (**A**) Representative brightfield images of *Xenopus* sperm from the control (DMSO) and benzotriazinone **10**-treated groups. Overlaid colored tracks represent the trajectories of individual spermatozoa, illustrating their motility patterns. (**B**) Immunofluorescence staining of sperm from the DMSO and benzotriazinone **10**-treated groups. Plasma membrane is labeled in red, α-tubulin (axonemal structure) in green, and nuclei counterstained with Hoechst 33342 in blue. DMSO-treated spermatozoa exhibited intact flagella, while benzotriazinone **10**-treated spermatozoa were deflagellated. Scale bar = 10 µm.

**Table 1 cells-14-00688-t001:** Compound numbers and chemical structures. Quantification of the bead track speed (µm/s) indicates the effect of each compound compared with the DMSO-treated embryos (*n* = 3 embryos, * *p* < 0.05, *** *p* < 0.001, ns, not significant).

Compound Number & Formula	Structure	Effect on Ciliary Function
**1** C_19_H_13_N_3_O	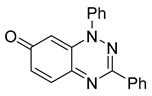	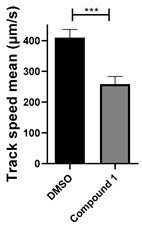
**2** C_14_H_8_F_3_N_3_O	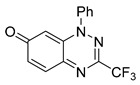	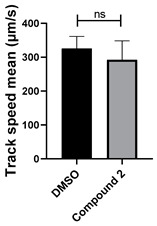
**3** C_19_H_14_N_4_O	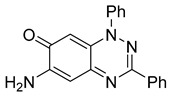	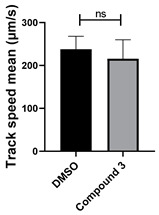
**4** C_21_H_18_N_4_O	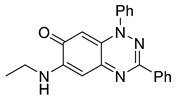	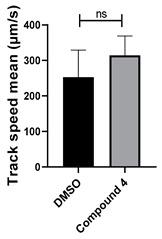
**5** C_25_H_24_N_4_O	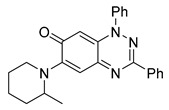	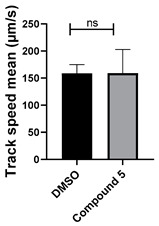
**6** C_25_H_24_N_4_O	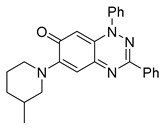	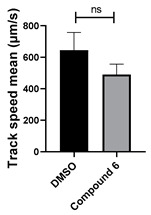
**7** C_25_H_24_N_4_O	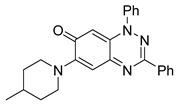	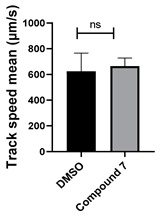
**8** C_23_H_20_N_4_O_2_	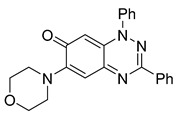	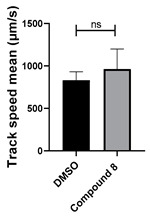
**9** C_24_H_23_N_5_O	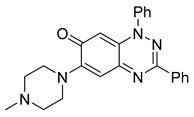	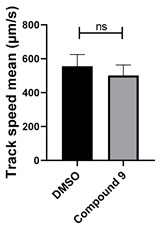
**10** C_29_H_25_N_5_O	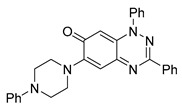	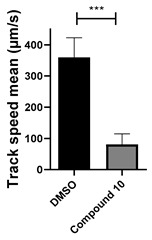
**11** C_30_H_26_N_4_O	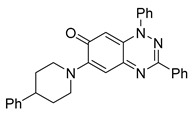	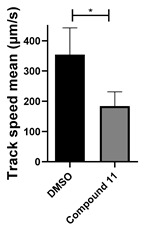
**12** C_25_H_19_N_5_O	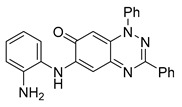	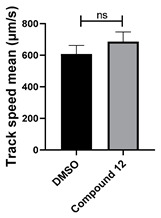
**13** C_25_H_27_N_5_O	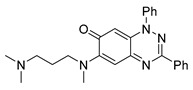	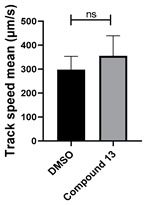
**14** C_20_H_15_N_3_O	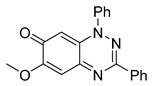	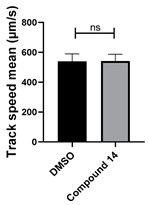
**15** C_25_H_17_N_3_OS	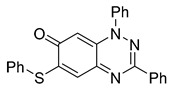	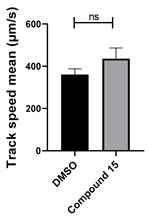
**16** C_23_H_15_N_3_OS	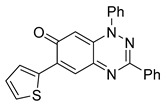	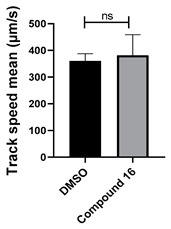
**17** C_23_H_18_N_4_O	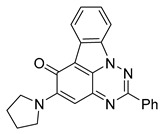	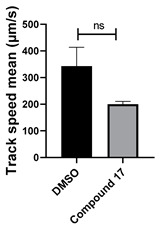
**18** C_25_H_15_N_3_OS	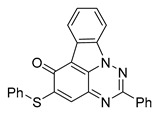	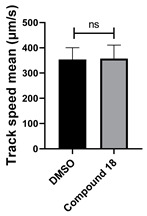
**19** C_25_H_17_N_3_O	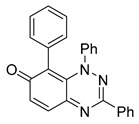	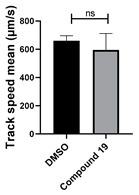
**20** C_26_H_19_N_3_O_2_	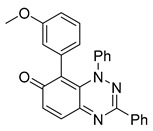	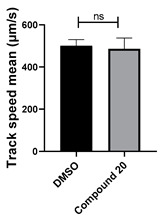
**21** C_25_H_16_ClN_3_O	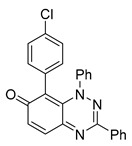	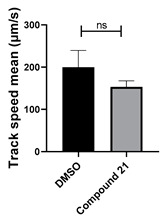
**22** C_23_H_15_N_3_O_2_	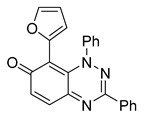	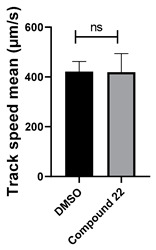

**Table 2 cells-14-00688-t002:** Antibodies used for immunofluorescence staining.

Antibody	Catalogue Number	Dilution
Acetylated alpha tubulin	Santa Cruz Biotechnology (Dallas, TX, USA), sc-23950	1:500
β-Tubulin (E7-s)	DSHB Hybridoma Products (Iowa City, IA, USA), AB2315513	1:500
Beta-catenin	Sino Biological (Chesterbrook, PA, USA), 11279-R021	1:250
Centrin 1	Proteintech (Rosemont, IL, USA), 12794-1-AP	1:500
CellMask plasma membrane stain	Invitrogen, C10046	1:1000
Phalloidin Alexa Fluor Plus 647	Thermo Fischer Scientific	1:500
Secondary donkey anti-rabbit Alexa 488	Invitrogen	1:500
Secondary donkey anti-mouse Alexa 488	Invitrogen	1:500
Secondary donkey anti-mouse Alexa 647	Invitrogen	1:500

## Data Availability

The datasets used and/or analyzed during the current study are available from the corresponding author upon request.

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
