# Peer review of "Reversible Modulation of Motile Cilia by a Benzo[e][1,2,4]triazinone: A Potential Non-Hormonal Approach to Male Contraception"

_cells, 2025, doi:10.3390/cells14100688_

Round 1

Reviewer 1 Report

Comments and Suggestions for Authors

1. In Line 230-231 the cilia regeneration began within 1 hour of drug removal, with complete recovery by 2 h. But in the Fig 2c, the significance between the DMSO group and 120s washout group was p < 0.05 (marked as  *). Why?

2. For time-lapse imaging, providing movies or full image coverage (e.g., co-staining with Hoechst and plasma membrane dyes) can enhance credibility.

3. On Line 290-297, please provide the correct Figure number.

4. More experiments will be needed to assess the safety and efficacy of benzotriazinone 10 in mammalian models (e.g., the drug’s efficacy in uterine ciliary cells). 

Author Response

  1. In Line 230-231 the cilia regeneration began within 1 hour of drug removal, with complete recovery by 2 h. But in the Fig 2c, the significance between the DMSO group and 120s washout group was p < 0.05 (marked as  *). Why?

Thank you for pointing this out. The statement in Line 230-231 about complete cilia recovery occurring by 2 hours was a typo. In fact, as shown in Fig. 2c, the cilia regeneration began within 1 hour of drug removal, with complete recovery by 3 hours. We apologize for this confusion and have corrected the text to reflect the accurate recovery time of 3 hours.

  1. For time-lapse imaging, providing movies or full image coverage (e.g., co-staining with Hoechst and plasma membrane dyes) can enhance credibility.

We appreciate this suggestion. To enhance the credibility and visual clarity of the time-lapse imaging, we have included movies in the supplementary material.

  1. On Line 290-297, please provide the correct Figure number.

Thank you for pointing this out. It is now corrected.

  1. More experiments will be needed to assess the safety and efficacy of benzotriazinone 10 in mammalian models (e.g., the drug’s efficacy in uterine ciliary cells). 

We agree with the reviewer that further studies in mammalian models are essential to fully assess the safety and efficacy of benzotriazinone 10. We have added this point to the conclusion section and acknowledge it as an important direction for future work.

Reviewer 2 Report

Comments and Suggestions for Authors

I really appreciate the effort the authors made to write such an interesting paper. However, I would like to address some minor issues:

  • Latin names like Xenopus, Xenopus laevis, in vitro should be consistently written in italic.
  • Section "2.7 Quantifications and statistics" is apparently missing.
  • Abbreviations should be explained when they first appear in text, e.g. "MCC" is explained at just sixth appearance in the text. Abbreviation "PBS" is not explained at all.
  • Both "[1,2,4]-triazin" and "[1,2,4]triazin" are written across the text, which must be unified.
  • Both "benzotriazinone 10-treated" and "benzotriazinone 10 treated" are written across the text, which mnust be unified.
  • There are many links to "Table 2", but there is not any table labeled as Table 2 (the table with chemical compounds is standing before Table 1).
  • In sections "2.1 General methods and materials", there are chemical substances labeled as (23) to (26), however, in the table there are only 22 chemical compounds. What are these numbers referring to?
  • Line 54: "as" should not be written in bold.
  • Line 129: Correct spelling of "Gentamicin" is "-mic-", not "-myc-".
  • Line 131: hCG is rather a "glycoprotein".
  • Line 132: "Petri dish" should be written with capital "P-".
  • Line 133: "dissected testes" - plural form is more appropriate.
  • Line 175: ... with with...
  • Line 381: "Hoechst" is name of the company, not a chemical substance.
  • Line 467: "Ca2+" should be written with superscript, same applies for "Fig. S4".
  • Fig. 4: A sentence is starting with small "b-".
  • Fig. S2: Typing error in "Benzotriazinone 10".
  • Fig. S5: Typing error in "Immunofluorescence".

Author Response

I really appreciate the effort the authors made to write such an interesting paper. However, I would like to address some minor issues:

  • Latin names like XenopusXenopus laevisin vitro should be consistently written in italic.
  • Section "2.7 Quantifications and statistics" is apparently missing.
  • Abbreviations should be explained when they first appear in text, e.g. "MCC" is explained at just sixth appearance in the text. Abbreviation "PBS" is not explained at all.
  • Both "[1,2,4]-triazin" and "[1,2,4]triazin" are written across the text, which must be unified.
  • Both "benzotriazinone 10-treated" and "benzotriazinone 10 treated" are written across the text, which mnust be unified.
  • There are many links to "Table 2", but there is not any table labeled as Table 2 (the table with chemical compounds is standing before Table 1).
  • In sections "2.1 General methods and materials", there are chemical substances labeled as (23) to (26), however, in the table there are only 22 chemical compounds. What are these numbers referring to?
  • Line 54: "as" should not be written in bold.
  • Line 129: Correct spelling of "Gentamicin" is "-mic-", not "-myc-".
  • Line 131: hCG is rather a "glycoprotein".
  • Line 132: "Petri dish" should be written with capital "P-".
  • Line 133: "dissected testes" - plural form is more appropriate.
  • Line 175: ... with with...
  • Line 381: "Hoechst" is name of the company, not a chemical substance.
  • Line 467: "Ca2+" should be written with superscript, same applies for "Fig. S4".
  • Fig. 4: A sentence is starting with small "b-".
  • Fig. S2: Typing error in "Benzotriazinone 10".
  • Fig. S5: Typing error in "Immunofluorescence".

We sincerely thank the reviewer for the kind feedback and for the careful review of the manuscript. We have addressed all the minor issues raised as follows:

  • All Latin names (Xenopus laevis, in vitro) have been italicized consistently. Typographic and capitalization issues (e.g., bolded "as" on line 54, lowercase "b-" in Fig. 4, "Petri dish" capitalization, and plural form “dissected testes”) have been corrected. Superscripts have been corrected for Ca²⁺.
  • All abbreviations, including MCC and PBS, are now defined at their first appearance. Chemical names have been standardized (e.g., “[1,2,4]-triazin”, “benzotriazinone 10-treated”).
  • Spelling errors (e.g., “Gentamycin” corrected to “Gentamicin”, "with with" on line 175, "Immunofluorescence" in Fig. S5) have been resolved.
  • The “Table 2” references were corrected following proper table renumbering.
  • Compounds (23)–(26) were included in the methods section because they were tested in the functional assay presented in Fig. 5A
  • The term “hCG” has been clarified as a glycoprotein hormone.
  • The reference to “Hoechst” has been revised to specify “Hoechst 33342 dye” to ensure accurate terminology.

We have carefully revised the manuscript to reflect these corrections. We are grateful for the reviewer’s attention to detail, which has helped improve the overall clarity and quality of the paper.

Reviewer 3 Report

Comments and Suggestions for Authors

In this manuscript  Chatzifrangkeskou and colleagues described the the effect of different benzotriazinyl derivatives on ciliary biology using the X. laevis as a biological model. 
Cilia are essential cell components related to many biological processes including fertilization. This work describes novel pharmacological tools to study cilia function with potential to be employed as a male contraceptive.  

The reported observations are based on a high-quality data and thus convincing. These results are highly important for the field of cell biology and reproduction as they can be potentially used as contraceptive.

I have only one major issue related to the writing of the manuscript. In the introduction the reader is introduced to the table number 2 which contains relevant results for the manuscript but is not clear why this table appears before the so-called Table 1 (methods section). As the authors progress with the results, relevant information is missing or not clearly mentioned. For example, authors claim "The highest concentration that allowed for embryo survival was selected for further testing (Fig. 1a)" however, no information regarding this concentration is mentioned in Fig 1a. neither in the figure legend. Then, the authors showed a series of immunolabeled cells, which are the core of their experiments, however the author failed in clarified several things:

  • In all results, microscopy images of MCCs cells are showed, however the definition of MCC is not declared until the discussion, furthermore, is not clear from which region of the embryo this MCCs were imaged.
  • information about the employed concentration is missing in figs. 2a,c, 3 and 4.
  • A very importante point is that there is NO information at all about the number of experiments performed or the statistical analysis used (indeed the section 2.7 is completely absent)
  • In fig. 3b, the authors may consider include some arrowheads to easily detect the severed cilia. 
  • I have a special concern about the results in Fig 4a. The lack of quantitative data makes it difficult the interpretation of this figure, since in the Benzotriazinone 10 treatment the label with acTub does not look different than the DMSO control. Maybe is just the DMSO chosen image.
  • Figure S3C legend does not match with the figure, actually, this is a copy paste of the legend in S2.
  • No scale bar shown in Fig 5a. In this figure the legend seems to be cropped. 

Another issue is, why there are a high variability (from 350 to 1000 um/s) in the track speed mean using DMSO in every shown experiment? 

In summary, the results are interesting and new. A major open and interesting question is the molecular mechanism of Benzotriazinone 10-mediated cilia shedding. 
